# On-surface synthesis of poly(p-phenylene ethynylene) molecular wires via in situ formation of carbon-carbon triple bond

Chen-Hui Shu[1], Meng-Xi Liu[2], Ze-Qi Zha[2,3], Jin-Liang Pan[2,3], Shao-Ze Zhang[1], Yu-Li Xie[1], Jian-Le Chen[1], Ding-Wang Yuan[4], Xiao-Hui Qiu[2,3] & Pei-Nian Liu[1]

The carbon–carbon triple bond (–C≡C–) is an elementary constituent for the construction of conjugated molecular wires and carbon allotropes such as carbyne and graphyne. Here we describe a general approach to in situ synthesize –C≡C– bond on Cu(111) surface via homo-coupling of the trichloromethyl groups, enabling the fabrication of individual and arrays of poly(p-phenylene ethynylene) molecular wires. Scanning tunneling spectroscopy reveals a delocalized electronic state extending along these molecular wires, whose structure is unraveled by atomically resolved images of scanning tunneling microscopy and noncontact atomic force microscopy. Combined with density functional theory calculations, we identify the intermediates formed in the sequential dechlorination process, including surface-bound benzyl, carbene, and carbyne radicals. Our method overcomes the limitation of previous on-surface syntheses of –C≡C– incorporated systems, which require the precursors containing alkyne group; it therefore allows for a more flexible design and fabrication of molecular architectures with tailored properties.

[1] Shanghai Key Laboratory of Functional Materials Chemistry, Key Laboratory for Advanced Materials, State Key Laboratory of Chemical Engineering and School of Chemistry and Molecular Engineering, East China University of Science and Technology, 130 Meilong Road, 200237 Shanghai, China. [2] CAS Key Laboratory of Standardization and Measurement for Nanotechnology, CAS Center for Excellence in Nanoscience, National Center for Nanoscience and Technology, 100190 Beijing, China. [3] University of Chinese Academy of Sciences, 100049 Beijing, China. [4] College of Materials Science and Engineering, Hunan University, 410082 Changsha, China. These authors contributed equally: Chen-Hui Shu, Meng-Xi Liu. Correspondence and requests for materials should be addressed to X.-H.Q. (email: xhqiu@nanoctr.cn) or to P.-N.L. (email: liupn@ecust.edu.cn)

The incorporation of carbon–carbon triple bond (–C≡C–) into π-conjugated systems is of great importance for the synthesis of carbon allotropes[1,2] and constructing molecular electronics[1,3–5], in which the electronically delocalized architectures are favorable for applications such as molecular wires, optical switches, transistors, and unimolecular rectifiers[6–10]. For conjugated oligomers or polymers, the rigidity inherent to –C≡C– bond can facilitate electron transfer through the organic backbone because of the enhanced electronic and electron–vibration coupling[11]. On the other hand, the high electron density associated with –C≡C– bond leads to high reactivity for versatile reactions, allowing the efficient preparation of polycyclic frameworks that further extend the functionality of the molecular systems[12,13].

The synthesis of –C≡C– bond is typically carried out in aqueous solution by coupling two trichloromethyl (–CCl₃) groups in the presence of a large excess of low-valent metal ions (such as $Cr^{2+}$ and $V^{2+}$ ions[14,15]) or two ethylidyne ligands of a trimolybdenum cluster[16]. The coupling reaction was initially proposed to occur via chromium–carbyne complexes[14]. Later on, using isotope-labeling experiments, the mechanism was indirectly proved to be the coupling between two free carbyne radicals, which were formed via a stepwise reduction of carbon–halide bonds, with metals acting as the reductants[15,17,18]. Despite the progress, solution synthesis of conjugated molecular wires remains extremely difficult due to the poor solubility of the polymerized products. Multiple long alkyl substituents are essential to enhance the solubility in this scenario[6–8], which nevertheless hinder further investigation of the intrinsic electronic properties of the targeted systems. Linear structure composed entirely of $sp$-hybridized carbon atoms was achieved inside carbon nanotubes using nanochemical reactions under high temperature and high vacuum conditions by taking advantage of the protection and confined environments of the nanoreactors[19], opening a new route to fabricate one-dimensional molecular wires.

On-surface synthesis becomes a powerful approach to fabricate well-controlled molecular nanostructures. In most reactions, organic monomers form C–C covalent bonds via Ullmann coupling reaction[20–22] and Glaser coupling reaction[23–26]. Only a few studies involved the formation of C=C bonds[27–30]. Due to the high reactivity of the –C≡C– bonds, in situ formation of –C≡C– bonds on metal surface has not been realized until very recently, when the acetylenic scaffolding has been synthesized on Au(111)[31]. Although ethynylene moieties can be introduced by Glaser coupling reaction, the rational design and controllable preparation of extended molecular structures containing ethynylene is still a great challenge and highly desired as a bottom-up approach to construct functional electronic components.

Herein, we report the in situ formation of a –C≡C– bond on Cu (111) via the coupling of two –CCl₃ groups of 1,4-bis(trichloromethyl)benzene (BTCMB) under ultra-high vacuum (UHV) conditions (Fig. 1a). Poly(p-phenylene ethynylene) (PPE) molecular wires are obtained at 300 K. The molecular structure of the product is characterized by scanning tunneling microscopy (STM) and noncontact atomic force microscopy (nc-AFM). The electronic state of PPE molecular wire is investigated by scanning tunneling spectroscopy (STS) and density functional theory (DFT) calculations. Moreover, the reaction mechanism is thoroughly examined using the combined approaches, which reveals the intermediate species formed in the sequential dechlorination steps.

## Results

**On-surface synthesis of PPE molecular wires on Cu(111).** In our approach, BTCMB was deposited onto a clean, single-crystal Cu(111) surface held at room temperature under UHV conditions (base pressure: $\sim 2 \times 10^{-10}$ mbar). The sample was then cooled to 4.7 K for further characterization. Large-scale STM image (Fig. 1b) shows spaghetti-like nanowires on the Cu(111)

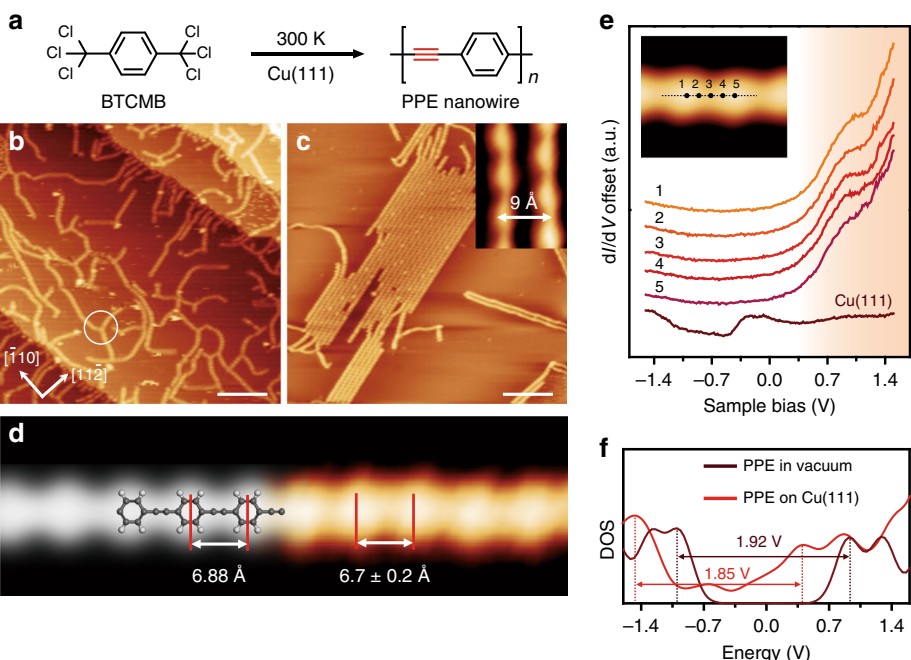

**Fig. 1** Formation of poly(p-phenylene ethynylene) (PPE) molecular wires on Cu(111). **a** Coupling of 1,4-bis(trichloromethyl)benzene (BTCMB) to form PPE molecular wires. **b** Scanning tunneling microscopy (STM) image after dosing BTCMB onto Cu(111) at 300 K ($I = 110$ pA, $V = -1.85$ V). Scale bar: 10 nm. **c** STM image of nanowires after annealing at 358 K ($I = 70$ pA, $V = -1.85$ V). Scale bar: 10 nm. **d** Simulated STM image (left part) and experimental STM image (right part, $I = 60$ pA, $V = -1.85$ V) of a PPE molecular wire, overlaid with the molecular model. Scale bar: 10 nm. **e** dI/dV spectra taken at different sites of the nanowire and Cu(111) surface. **f** Comparison of calculated density of states (DOS) for PPE in vacuum and on Cu(111) surface

surface. The ends of most nanowires were attached to Cu step edges. Branched structures were also observed, as marked by white circle in Fig. 1b. We attributed the three-fold nodes to a coordination structure consisting of three nanowires attached to a Cu adatom at the joint point[32,33] (see Supplementary Fig. 1 for detailed characterizations and DFT calculations). Further annealing to 358 K resulted in raft-like nanowire arrays, which aligned to directions deviating by ± 19° from $[0\bar{1}1]$ or the equivalent orientations of Cu(111) (Fig. 1c and Supplementary Fig. 2). The nanowires grew longer after an extended annealing at 358 K, which indicated that the nanowires with termini attached to step edges or Cu adatoms coupled with each other to form longer nanowires. The remaining nanowire termini were passivated by hydrogen atoms, thereby forming –CH$_3$ (Supplementary Fig. 3) after annealing at 358 K. The elongated nanowires could cross the Cu steps with the geometric order remaining unchanged (Supplementary Fig. 4). The close-up STM image (inset in Fig. 1c) shows that the distance between the adjacent nanowires is 9 Å. The nanowire shows periodic features with a pitch of 6.7 ± 0.2 Å between adjacent protrosions (right part of Fig. 1d). This observation agrees with our simulated STM image of PPE molecular wire, which shows a similar periodicity of 6.88 Å corresponding to the unit of phenylene and ethynylene (left part of Fig. 1d). A careful comparison of the experimental and calculated results suggests that the protrosions and the linkers in the STM image can be assigned to phenylene and ethynylene, respectively.

STS measurements were carried out to investigate the electronic state of the nanowire. In contrast to the Shockley-type surface state at approximately −0.4 V of Cu(111), the d$I$/d$V$ spectra acquired at various positions (marked in the inset of Fig. 1e) along the nanowire show a prominent state with an onset at 0.5 V and a shoulder peaked at 1.1 V (Fig. 1e). The high similarity of the spectra acquired on different sites of the nanowire indicates that the electronic state is delocalized over the nanowire. The calculated density of states (DOS) of PPE

molecular wire predicts a bandgap of 1.92 eV in vacuum and 1.85 eV on Cu(111) (Fig. 1f and Supplementary Fig. 5). The hybridization of electronic state of PPE with that of the underlying Cu substrate leads to the reduction of bandgap by ~70 meV and the downshift of bandedges of the molecular wire. Projected DOS (PDOS) of PPE molecular wire on Cu(111) reveals that the hybridization is mainly attributed to the interaction between $p_z$ orbitals of C atoms in PPE nanowire and $d_z^2$ orbitals of Cu substrates (Supplementary Fig. 6e, f ). Compared to the calculations, the detected state in the d$I$/d$V$ spectra rose from 0.5 eV and can be assigned to the conduction band of nanowire, although the valence band was not observed in d$I$/d$V$ spectra. For a similar case of graphene nanoribbon on Au(111), it suggested that the valence band decayed rapidly with increasing tip-sample distance and could be resolved only at set point currents >10 nA[34]. In our work, the valence band was not observed, likely because it was too weak relative to the strong contribution from the underlying Cu(111)[34–36].

In general, the –C≡C– bond has a very high reactivity on Cu surface. For instance, the –C≡C– bond in acetylene hybridized with Cu substrate and was lengthened by 0.21(2) Å, resulting in a bent geometry of acetylene[37,38]. Therefore, on-surface synthesis of acetylenic scaffoldings so far has been reported mostly on Au and Ag[23,24,39] and rarely on Cu. As shown in Fig. 1, the mobile nanowires with delocalized electronic state on Cu(111) indicated that –C≡C– bond did not bound with the substrate. Our calculations reveal that the bond order of the –C≡C– triple bonds in PPE nanowire is 2.61 on Cu(111) surface, which is very close to that in vacuum (2.62). The electron localization function maps and charge density difference calculations indicate insignificant electronic hybridization between PPE nanowire and the underlying Cu substrate (Supplementary Fig. 6b–d), as also confirmed by Bader charge analysis, which shows a small amount of charge transfer from Cu(111) to PPE nanowire by 0.03e per unit (–Ar-C≡C–).

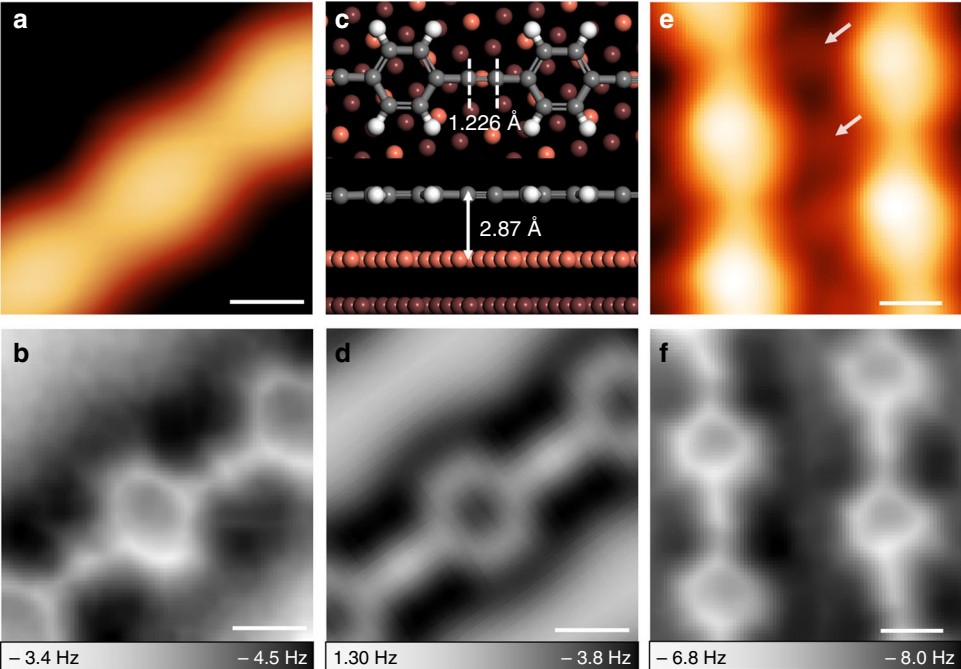

**Fig. 2** Structural characterization of PPE molecular wire. **a** STM image ($I$ = 20 pA, $V$ = −0.6 V) and the corresponding **b** noncontact atomic force microscopy (nc-AFM) image of a nanowire acquired with a CO-functionalized tip. **c** Density functional theory (DFT)-optimized structure of oligomeric PPE on Cu(111). **d** Simulated nc-AFM image based on the molecular model in **c**. **e** STM image and **f** the corresponding nc-AFM image of a segment of raft-like PPE nanowire arrays and the embedded Cl atoms (constant-height mode, $V$ = 2 mV). Scale bars (**a**, **b**, **d**-**f**): 300 pm

Figure 2a, b are the high-resolution STM image and corresponding nc-AFM image of a section of the nanowire. The phenylene moieties are clearly resolved in the nc-AFM image, and they correspond to the protrusions in the STM image. Enhanced contrast can be observed in the middle of the linkage between two adjacent phenylenes, which is attributed to the large electron localization in the region of –C≡C– bonds (Supplementary Fig. 6b)[12]. The successful formation of –C≡C– on the Cu(111) surface is also confirmed by the linear bond configuration of the two *sp*-hybridized carbon atoms. We performed DFT calculations of PPE molecular wire adsorbed on Cu(111). As shown in Fig. 2c, the optimized ethynylene maintains a linear configuration, and the calculated –C≡C– bond length is 1.226 Å, which is very close to the bond length calculated in vacuum (1.219 Å). AFM simulation (Fig. 2d) based on this model agrees well with the nc-AFM image in Fig. 2b. As a result, we believe that highly selective coupling of BTCMB produced PPE molecular wires on Cu(111). The distortion along the direction perpendicular to the axis of PPE wire observed both in AFM image and simulation was induced by the tilting effect of CO on tip apex[40].

We acquired the close-up STM image and nc-AFM image of close-packed PPE nanowire array. Different from the STM image in inset of Fig. 1c acquired under large bias (−1.85 V), dot-like protrusions embedded between two neighboring PPE nanowires were observed in Fig. 2e (2 mV). AFM image (Fig. 2f ) revealed precise molecular structure of PPE nanowire arrays with an axial displacement between neighboring wires. We believed that the dot-like protrusions were Cl atoms detached from precursors typically seen in the coupling reactions using Cl-substituted precursors[41]. DFT calculation (Supplementary Fig. 7) indicated that the Cl atoms stabilized PPE nanowire arrays by inter-molecular Cl…H bonds[41,42].

**Mechanism of on-surface –C≡C– bond formation.** To understand the reaction mechanism, we used TCMB as the model system (Fig. 3a) and conducted sequential annealing experiment. TCMB has only one –CCl₃ group making the reaction process easily be tracked by STM and AFM. For TCMB deposited at room temperature, dumbbell-shaped molecules orienting along the [0$\bar{1}$1] or equivalent directions of Cu(111) were observed surrounded by chlorine atoms (Fig. 3b). After annealing the sample at 358 K, the dumbbell-shaped molecules self-assembled to ordered structure (Fig. 3c). High-resolution STM and corresponding nc-AFM images (Fig. 3d, e) of an individual molecule confirmed that the dumbbell-shaped molecules were the 1,2-diphenylethyne (DPE).

The above chemical transformation was closely examined using a sequential annealing technique. TCMB was deposited on Cu(111) at 4.7 K and we then gradually elevated the substrate temperature to obtain the reaction intermediates. A careful statistical analysis of large-scale STM images and a number of AFM images of surface-stabilized species (Supplementary Figs. 8, 9) was performed. STM and the corresponding nc-AFM images of the as-deposited molecule (Fig. 4a, d) revealed a tilted phenyl and two adjacent protrusions. AFM simulation and DFT calculation (Fig. 4g, j) suggest that the observed molecule is the benzyl radical I, formed by dechlorination of TCMB and bound with the underlying Cu(111). The identification of I using AFM images was carefully described in Supplementary Fig. 8. We have counted the molecular species on Cu(111) formed at 4.7 K and nearly 90% of them were I (Supplementary Fig. 9l). Upon annealing at ~50 K, the recorded molecule (Fig. 4b, e) showed a different feature from I, which contained a slightly tilted phenyl and only one chlorine atom. Calculated results (Fig. 4h, k) suggested that the molecule is the surface-bound carbene radical II, which agreed well with nc-AFM image (Fig. 4e). Nearly 70% of the species after annealing at

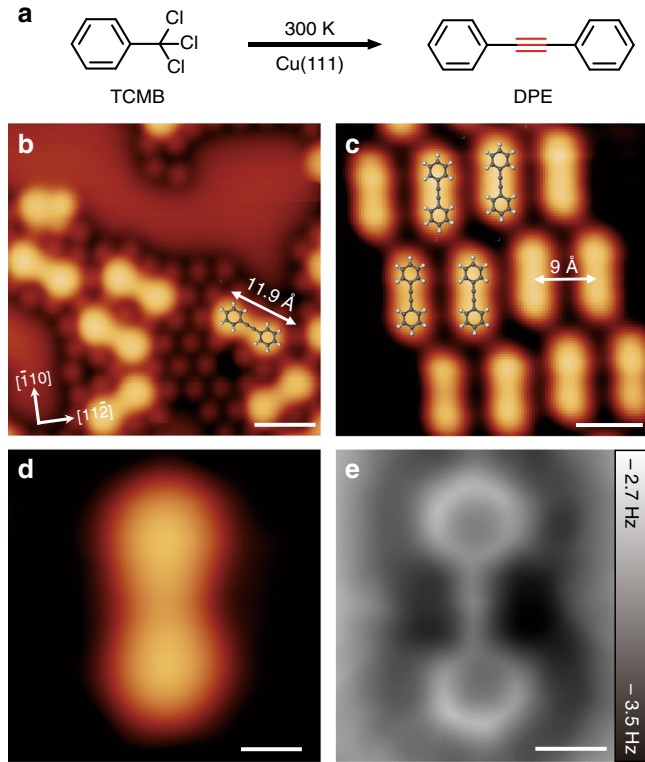

**Fig. 3** Coupling reaction of trichloromethylbenzene (TCMB) on Cu(111). **a** Coupling of TCMB to form 1,2-diphenylethyne (DPE). **b** STM image after TCMB deposition onto a Cu(111) surface at 300 K (*I* = 20 pA, *V* = −0.6 V). Scale bar: 1 nm. **c** STM image of close-packed DPE molecules after annealing to 358 K (overlaid with molecular models). The molecular length and intermolecular distance is 11.9 Å and 9 Å, respectively (*I* = 20 pA, *V* = −0.6 V). Scale bar: 1 nm. **d** High-resolution STM image (*I* = 20 pA, *V* = −0.6 V, scale bar: 300 pm) and **e** the corresponding nc-AFM image of DPE molecules. Scale bar: 300 pm

50 K were II, while the remaining 30% were I (Supplementary Fig. 9l). Further annealing to ~70 K resulted in a new species that there was no molecular skeleton shown in STM and nc-AFM images (Fig. 4c, f ). The height of this species was measured by nc-AFM of ~7 Å (Supplementary Fig. 10)[43]. DFT calculation reveals that carbyne radical III (formed by dechlorinating three chlorine atoms from TCMB) prefers to adsorb on three-fold hollow site of Cu(111), and the surface-stabilized carbyne radical shows a height of 6.7 Å relative to Cu surface (Fig. 4l). The experimental and simulated nc-AFM images are consistent, confirming the generation of III (Fig. 4f, i). Finally, the sample was annealed to room temperature for 30 min and re-cooled to 4.7 K. The DPE molecules can be observed on Cu(111) (Supplementary Fig. 11). The reaction intermediates we proposed could be rationalized by the homocoupling of 1, 1, 1-trichloromethylaryl in solution, which yielded the same species that have been well characterized by means of spectroscopy[15]. We also detected molecular islands formed by self-assembled structures of I or II on Cu(111) as shown in Supplementary Fig. 9; however, homocoupling products of I or II, as well as other by-products, were not detected during this elevated annealing experiment.

The mechanism for the coupling reaction can be concluded as follows (Fig. 5c): (1) TCMB molecule dechlorinated on Cu(111) and sequentially generated the surface-bound benzyl, carbene, and carbyne radicals following the elevated annealing process; (2) two carbyne radicals diffused on Cu(111) and coupled to form DPE. To clarify the reaction pathway, we calculated the energy

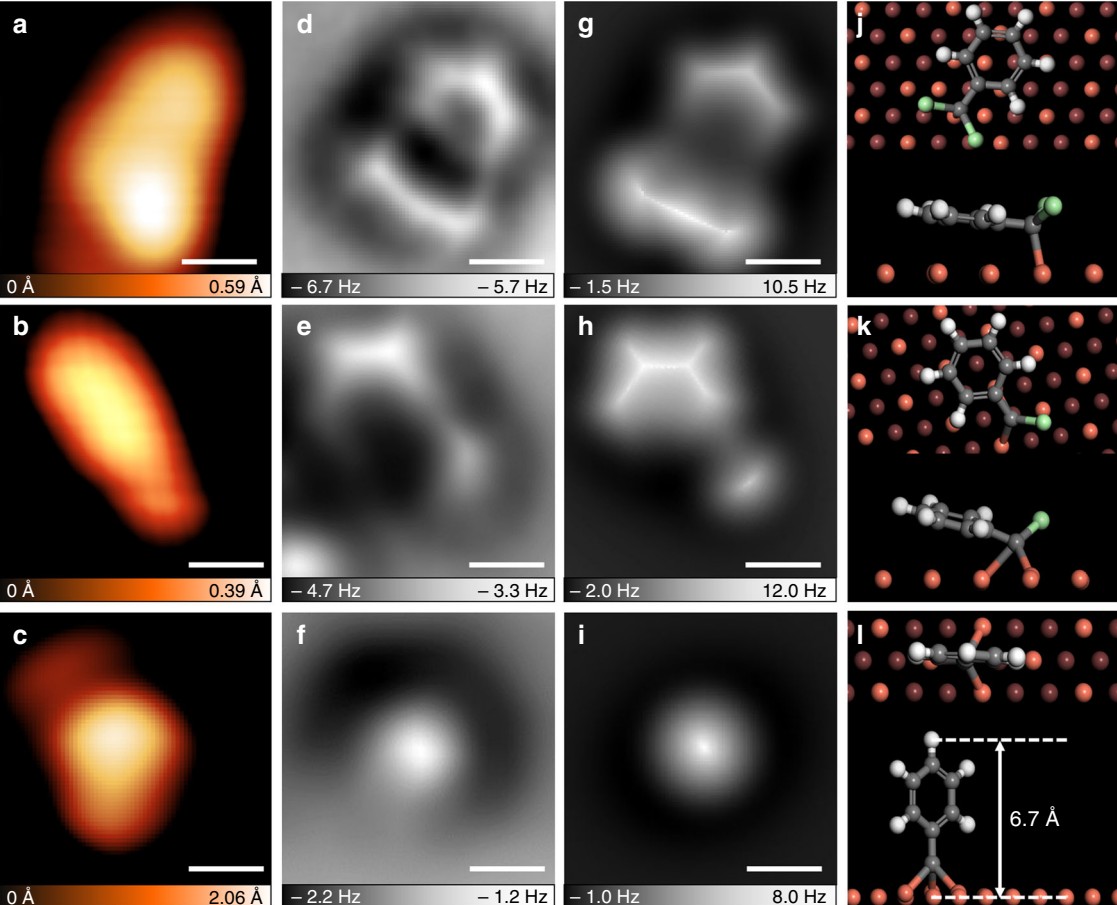

**Fig. 4** Direct imaging of reaction intermediates of TCMB coupling on Cu(111). **a** STM image of the intermediate I observed after TCMB deposited on Cu(111) at 4.7 K ($I = 20$ pA, $V = -0.6$ V). **b**, **c** STM images of intermediates II and III observed after elevated annealing process ($I = 20$ pA, $V = -0.6$ V). **d–f** The corrreponding nc-AFM images of **a–c**. **g–i** Simulated AFM images of the three typical reaction intermediates in **d–f**. **j–l** Top and side views of DFT-optimized models of the typical intermediates. Scale bars (**a–i**): 300 pm

landscape of the reaction as illustrated in Fig. 5a, b. We found that the reaction barrier for dechlorination process from TCMB to benzyl radical intermediate I, I to carbene radical intermediate II, II to carbyne radical intermediate III are gradually increased (0.01, 0.10, and 0.29 eV), which is in good agreement with the elevated reaction temperature. In comparison, reaction barriers for the sequentially dechlorination steps in gas phase are 2.24, 4.02, and 4.04 eV. The significant difference indicates that Cu (111) surface has a strong catalytic effect on dechlorination reaction by reducing the reaction barriers, thereby reducing the experimetal reaction temperature. In addition, the sequential dechlorination process of TCMB on Cu(111) is exothermic with reaction energies of $-2.07$, $-1.25$, and $-1.33$ eV, which indicates that the formation of carbyne radical intermediate III is energetically favorable. The subsequent formation of $-C\equiv C-$ bond is governed by the coupling reaction between surface-bound intermediates. The reaction barriers for the homocoupling of I, II, and III were calculated to be 2.16, 0.70, and 0.47 eV (Supplementary Fig. 12 and Fig. 5b), respectively. We found that the first two reaction barriers (homocoupling of I and II) are much higher than that of the dechlorination of I and II; therefore, the surface-bound intermediates I and II prefer to be dechlorinated rather than be homocoupled. Therefore, we can exclude the reaction pathways of homocoupling of benzyl radical intermediate I, as well as carbene radical intermediate II. Putting together the calculated results of sequential dechlorinations of TCMB and homocoupling of radical intermediates, the gradually increased

reaction barriers ensure the high selectivity of $-C\equiv C-$ formation, and the last step (homocoupling of III) is expected to be the rate-determining step due to the largest reaction barrier.

In general, free carbyne radicals are the intermediates for $-C\equiv C-$ formation in solution[15,17,18]. The extremely active free carbyne radicals reacted with each other as well as the solvent molecules, generating various by-products and leading to poor selectivity of coupling reaction[15]. In contrast, as proved by our experimental and calculated results, the Cu(111) surface plays a significant role in this coupling reaction, not only catalyzing the coupling reaction but also acting as a supporting platform by stabilizing the radical intermediates and constraining the nanostructure into two dimensions. We found that the PPE nanowires can also be successfully synthesized on Ag(111) (Supplementary Fig. 13). This on-surface approach overcomes the limitation of poor solubility for nanowire synthesized in solution, as well as kept the extended π-conjugation along the molecular wire by retaining the phenyl rings in the same plane, which ensures further investigation of intrinsic electronic property of such molecular wires.

## Discussion

We reported the on-surface synthesis of PPE molecular wires on Cu(111) via a homocoupling reaction of $-CCl_3$ functional groups of commercially available precursor BTCMB. STS measurement indicated that the molecular electronic state was delocalized along

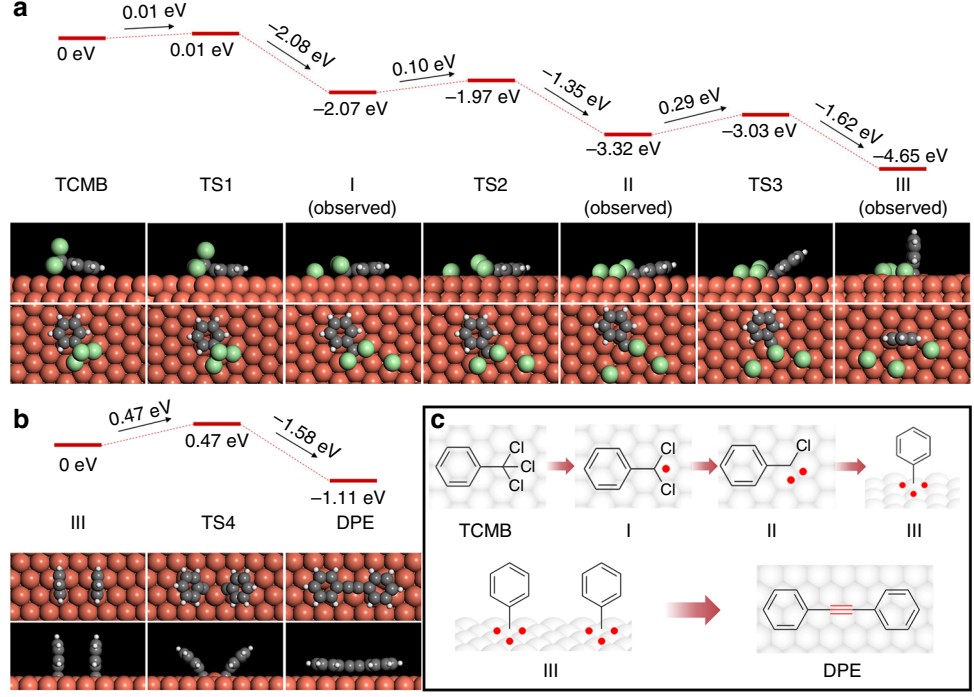

**Fig. 5** Reaction parthway for the homocoupling of TCMB on Cu(111). **a** Calculated energy and molecular structures for intermediates (observed) and transition states of the dechlorination process of TCMB on Cu(111). **b** Calculated energy of coupling process of III and the formation of DPE. **c** Overview of the reaction pathway for the formation of DPE on Cu(111)

the molecular wire because of the weak interaction between –C≡C– bonds with the underlying metallic substrate. The reaction mechanism was investigated using STM, nc-AFM in combination with DFT calculations, and revealed the sequential dechlorination steps involving surface-bound intermediates of benzyl, carbene, and carbyne radicals, which coupled to form –C≡C– bond. The presented protocol for in situ synthesizing –C≡C– functional group on the surface overcomes the limitation of using precursors containing alkyne group in the previous work, and therefore offers more flexibility in design and further exploration of tailored molecular architectures.

## Methods

**Experimental measurement.** Single crystalline Cu (111) surface was cleaned by cycles of Ar$^+$ sputtering and annealing under UHV (base pressure $2 \times 10^{-10}$ mbar). BTCMB and TCMB were purchased from TCI company. BTCMB and TCMB molecules were dosed onto Cu(111) through leakage valve. STM measurements were performed on an Omicron low-temperature STM operated at 4.7 K with an electrochemically etched tungsten tip. The STM images were taken in the constant-current mode and the voltages refer to the bias on samples with respect to the tip. The d$I$/d$V$ spectra were acquired by a lock-in amplifier, while the sample bias was modulated by a 553 Hz, 30 mV (r.m.s.) sinusoidal signal under open-feedback conditions. The tip state was checked via the appearance of the characteristic Shockley-type surface state on clean Cu(111) surfaces. The nc-AFM measurement was carried out at LHe temperature in constant-height frequency modulation mode with a CO-functionalized tip (resonance frequency $f_0 \approx 40.7$ kHz, oscillation amplitude A ≈ 100 pm, quality factor $Q \approx 5.6 \times 10^4$).

**Theoretical calculation.** DFT calculations were performed with the VASP code[44,45], using the projector-augmented wave method[46,47]. We used the generalized gradient approximation with Perdew–Burke–Ernzerhof formulism to treat exchange–correlation interaction[48], and van der Waals interactions were considered by using the DFT-D3 developed by Grimme[49]. The simulated AFM images were performed using the model provided by Hapala et al.[50] with a flexible CO tip ($k_{tip} = 0.5$ N m$^{-1}$).

**Data availability.** The authors declare that the main data supporting the findings of this study are available within the paper and its Supplementary Information files. Extra data are available from the corresponding authors upon reasonable request.

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

# ARTICLE

15. Levy, O. & Bino, A. Metal ions do not play a direct role in the formation of carbon–carbon triple bonds during reduction of trihaloalkyls by $Cr^{II}$ or $V^{II}$. *Chem. Eur. J.* **18**, 15944–15947 (2012).

16. Bino, A., Ardon, M. & Shirman, E. Formation of a carbon–carbon triple bond by coupling reactions in aqueous solution. *Science* **308**, 234–235 (2005).

17. Bogoslavsky, B. et al. Do carbyne radicals really exist in aqueous solution? *Angew. Chem. Int. Ed.* **51**, 90–94 (2012).

18. Danovich, D. et al. Formation of carbon–carbon triply bonded molecules from two free carbyne radicals via a conical intersection. *J. Phys. Chem. Lett.* **4**, 58–64 (2013).

19. Shi, L. et al. Confined linear carbon chains as a route to bulk carbyne. *Nat. Mater.* **15**, 634–639 (2016).

20. Dong, L., Liu, P. N. & Lin, N. Surface-activated coupling reactions confined on a surface. *Acc. Chem. Res.* **48**, 2765–2774 (2015).

21. Grill, L. et al. Nano-architectures by covalent assembly of molecular building blocks. *Nat. Nanotechnol.* **2**, 687–691 (2007).

22. Fan, Q., Gottfried, J. M. & Zhu, J. Surface-catalyzed C–C covalent coupling strategies toward the synthesis of low-dimensional carbon-based nanostructures. *Acc. Chem. Res.* **48**, 2484–2494 (2015).

23. Zhang, Y. Q. et al. Homo-coupling of terminal alkynes on a noble metal surface. *Nat. Commun.* **3**, 1286 (2012).

24. Gao, H.-Y. et al. Glaser coupling at metal surfaces. *Angew. Chem. Int. Ed.* **52**, 4024–4028 (2013).

25. Klappenberger, F. et al. On-surface synthesis of carbon-based scaffolds and nanomaterials using terminal alkynes. *Acc. Chem. Res.* **48**, 2140–2150 (2015).

26. Klappenberger, F. et al. Functionalized graphdiyne nanowires: on-surface synthesis and assessment of band structure, flexibility, and information storage potential. *Small* **14**, 1704321 (2018).

27. Matena, M. et al. Transforming surface coordination polymers into covalent surface polymers: linked polycondensed aromatics through oligomerization of N-heterocyclic carbene intermediates. *Angew. Chem. Int. Ed.* **47**, 2414–2417 (2008).

28. Kalashnyk, N. et al. On-surface synthesis of aligned functional nanoribbons monitored by scanning tunnelling microscopy and vibrational spectroscopy. *Nat. Commun.* **8**, 14735 (2017).

29. Sun, Q. et al. On-surface formation of cumulene by dehalogenative homocoupling of alkenyl gem-dibromides. *Angew. Chem. Int. Ed.* **129**, 12333–12337 (2017).

30. Ruffieux, P. et al. On-surface synthesis of graphene nanoribbons with zigzag edge topology. *Nature* **531**, 489–492 (2016).

31. Sun, Q. et al. Direct formation of C–C triple-bonded structural motifs by on-surface dehalogenative homocouplings of tribromomethyl-substituted arenes. *Angew. Chem. Int. Ed.* **130**, 4099–4102 (2018).

32. Gutzler, R. et al. Surface mediated synthesis of 2D covalent organic frameworks: 1,3,5-tris(4-bromophenyl)benzene on graphite(001), Cu(111), and Ag(110). *Chem. Commun.* **45**, 4456–4458 (2009).

33. Zhang, Y. Q. et al. Unusual deprotonated alkynyl hydrogen bonding in metal-supported hydrocarbon assembly. *J. Phys. Chem. C* **119**, 9669–9679 (2015).

34. Talirz, L. et al. On-surface synthesis and characterization of 9-atom wide armchair graphene nanoribbons. *ACS Nano* **11**, 1380–1388 (2017).

35. Kawai, S. et al. Atomically controlled substitutional boron-doping of graphene nanoribbons. *Nat. Commun.* **6**, 8098 (2015).

36. Wang, S. et al. Resolving band-structure evolution and defect-induced states of single conjugated oligomers by scanning tunneling microscopy and tight-binding calculations. *Phys. Rev. Lett.* **106**, 206803 (2011).

37. Stipe, B. C., Rezaei, M. A. & Ho, W. Coupling of vibrational excitation to the rotational motion of a single adsorbed molecule. *Phys. Rev. Lett.* **81**, 1263–1266 (1998).

38. Arvanitis, D., Döbler, U., Wenzel, L., Baberschke, K. & Stöhr, J. Position of the $\sigma$-shape and $\pi$ resonances of $C_2H_2$, $C_2H_4$ and $C_2H_6$ on Cu(100) at 60 K: A NEXAFS study. *Surf. Sci.* **178**, 686–692 (1986).

39. Sun, Q. et al. Dehalogenative homocoupling of terminal alkynyl bromides on Au(111): incorporation of acetylenic scaffolding into surface nanostructures. *ACS Nano* **10**, 7023–7030 (2016).

40. Moll, N. et al. Image distortions of a partially fluorinated hydrocarbon molecule in atomic force microscopy with carbon monoxide terminated tips. *Nano. Lett.* **14**, 6127–6131 (2014).

41. Shi, K. J. et al. Ullmann reaction of aryl chlorides on various surfaces and the application in stepwise growth of 2D covalent organic frameworks. *Org. Lett.* **18**, 1282–1285 (2016).

42. Park, J. et al. Interchain interactions mediated by Br adsorbates in arrays of metal-organic hybrid chains on Ag(111). *J. Phys. Chem. C* **115**, 14834–14838 (2011).

43. Schuler, B. et al. Adsorption geometry determination of single molecules by atomic force microscopy. *Phys. Rev. Lett.* **111**, 106103 (2013).

44. Kresse, G. & Hafner, J. Ab initio molecular dynamics for open-shell transition metals. *Phys. Rev. B* **48**, 13115–13118 (1993).

45. Kresse, G. & Furthmüller, J. Efficient iterative schemes for ab initio total-energy calculations using a plane-wave basis set. *Phys. Rev. B* **54**, 11169–11186 (1996).

46. Kresse, G. & Joubert, D. From ultrasoft pseudopotentials to the projector augmented-wave method. *Phys. Rev. B* **59**, 1758–1775 (1999).

47. Blöchl, P. E. Projector augmented-wave method. *Phys. Rev. B* **50**, 17953–17979 (1994).

48. Perdew, J. P., Burke, K. & Ernzerhof, M. Generalized gradient approximation made simple. *Phys. Rev. Lett.* **77**, 3865–3868 (1996).

49. Grimme, S. Semiempirical GGA-type density functional constructed with a long-range dispersion correction. *J. Comput. Chem.* **27**, 1787–1799 (2006).

50. Hapala, P., Temirov, R., Tautz, F. S. & Jelínek, P. Origin of high-resolution IETS-STM images of organic molecules with functionalized tips. *Phys. Rev. Lett.* **113**, 226101 (2014).

## Acknowledgements

This work was supported by the National Natural Science Foundation of China (Project Nos. 21421004, 21425310, 21561162003, 21603045, and 21672059), the Program for Eastern Scholar Distinguished Professor, the Programme of Introducing Talents of Discipline to Universities (B16017), the Ministry of Science and Technology of China (Grant No. 2016YFA0200700, 2017YFA0205000), and CAS-PKU Pioneer Cooperation Team.

## Author contributions

C.H.-S., M.-X.L., Z.-Q.Z., J.-L.P., and Y.-L.X. conducted experiments. C.H.-S., S.-Z.Z. D.-W.Y., and J.-L.C. conducted theoretical computations. P.-N.L. conceived and planned the project. P.-N.L. and X.-H.Q. carried out data analysis and supervised the project. All the authors discussed the results and helped in writing the manuscript.

## Additional information

**Competing interests:** The authors declare no competing interests.

