## [Peer Review File · Nature Communications]

Reviewers' comments:

Reviewer #1 (Remarks to the Author):

This is a very interesting and timely manuscript. Interest in surface polymerization is at an all time high and the authors present a novel chemistry for creating conducting wires. The list of citations and their motivation for the study are appropriate and clear. The English is fairly good with only a few rough spots (e.g. appeared no distinct). The quality of the data appears to be very high. My only minor recommendation is that they explain why there are long rafts of wires rather than a jumble of varying lengths with gaps and mixed orientation in the three equivalent symmetry directions.

This paper will clearly be of great general interest and will be highly cited. I recommend publication after the authors consider my comments. I do not need to re-review.

Reviewer #2 (Remarks to the Author):

In this manuscript, the authors report on carbon-carbon triple bond formations on a Cu(111) surface via surface-assisted dechlorination of trichloromethyl groups and concomitant coupling between carbyne radicals. The main conclusion is supported by a combinatorial investigation via STM/STS and nc-AFM of two different starting precursors, yielding respectively PPE wires and DPE dimers. The nc-AFM characteristics of the formed alkyne unit agrees well with earlier reports. Furthermore, a plausible reaction pathway is proposed based on the STM/AFM examination of the reaction intermediates as well as DFT modeling.

The findings and elegant approach introduce an important coupling reaction to the growing field of on-surface chemistry, which opens new avenues towards the synthesis of low-dimensional materials incorporating sp+sp² hybridized carbons without employing alkyne derivatives. The ms is overall well written, is based on high-quality experimental data, an insightful analysis and touches a timely topic.

Prior to publication a series of points deserves further attention:

- regarding carbyne synthesis (see introduction), it is interesting to acknowledge the advances reported in Nature Materials 15, 634–639 (2016) that also take advantage of special environments and reaction conditions
- it was pointed out that 'Glaser coupling' is not (always) adequate for interfacial reaction scenarios.
- the nature of three-fold node of the branched structure formed at 300 K (Fig.1b and Fig. S1) is not addressed. Similar to the fact that most of the ends of the molecular chains anchor to the step edges, the presence of a Cu atom in the joint connection could indicate the stabilization of carbon radicals via metal atoms (cf. J. Phys. Chem. C 119, 9669 (2015)). The authors shall discuss the possible node structures.
- likewise, what is the chemical structure of the PPE wire termini after 358 K annealing, especially for those resting on the open surface (Fig. 1C)? Is there evidence for surface-stabilized radicals or their passivation by hydrogen atoms? Please include meaningful discussions in the main text.
- in the inset of Fig.1c, there is a vertical displacement between the neighboring wires about half of its periodicity that should be analysed.

- it is helpful to show the gas-phase band structure of the PPE wire to have a better understanding of the DOS profile depicted in Fig. 1f.
- the data in fig. 1 also indicate that atomic steps of the substrate can be crossed by reaction products without affecting the geometric order. Indeed, it would be interesting to discuss potential templating routes towards improved control of molecular wires, as demonstrated in the literature.
- p. 5: although AFM data and simulations roughly agree, significant distortions seem to affect the experimental data. The reasons should be clarified.
- p. 5: the carbon-carbon bonding description and (potential) substrate hybridization effects are described in an unconvincing manner. This crucial issue requires a more rigorous treatment.
- p. 8: the identification of reaction intermediates is partially rather speculative. Although certain similarities between simulated and experimental AFM data exist, the agreement is not generally very good. Discrepancies and potential reasons should be adequately addressed.
- the experimental findings to support the proposed reaction pathway are not properly presented. When referring to the reaction intermediates, large-scale STM overview images of samples subject to different annealing temperatures are not shown. Statistics of the distribution of intermediates formed at different temperatures shall be provided. Alternatively, a stepwise tip-induced dechlorination experiment can be performed to verify the geometrical changes of the precursor upon activation.
- is it expected that this reaction scheme also would work on noble metal or other surfaces? This is an important point regarding broad employability of the approach.
- there are repeated spelling / grammatical mistakes, careful proofreading is necessary.

Reply to reviewers' comments

We sincerely thank the referees for their careful reading and suggestions for improving our paper. We have revised the manuscript on the basis of these comments. The detailed responses are attached below.

Reviewer #1:

This is a very interesting and timely manuscript. Interest in surface polymerization is at an all-time high and the authors present a novel chemistry for creating conducting wires. The list of citations and their motivation for the study are appropriate and clear. The English is fairly good with only a few rough spots (e.g. appeared no distinct). The quality of the data appears to be very high. My only minor recommendation is that they explain why there are long rafts of wires rather than a jumble of varying lengths with gaps and mixed orientation in the three equivalent symmetry directions.

This paper will clearly be of great general interest and will be highly cited. I recommend publication after the authors consider my comments. I do not need to re-review.

Reply: We appreciate the referee's constructive comments. After annealing at 358 K, PPE nanowires self-assembled into raft-like arrays, as illustrated in the large-scale STM image below. As the reviewer pointed out, the raft-like PPE nanowire arrays have varying lengths and orientations rotating by $\pm 19^\circ$ from $[01\bar{1}]$ or the equivalent orientations of Cu(111). We added this STM image and discussion as supplementary materials (Fig. S2) in the revised manuscript.

Figure S2 STM image of raft-like PPE nanowire arrays after annealing at 358 K.

Reviewer #2:

In this manuscript, the authors report on carbon-carbon triple bond formations on a Cu(111) surface via surface-assisted dechlorination of trichloromethyl groups and concomitant coupling between carbyne radicals. The main conclusion is supported by a combinatory investigation via STM/STS and nc-AFM of two different starting precursors, yielding respectively PPE wires and DPE dimers. The nc-AFM characteristics of the formed alkyne unit agrees well with earlier reports. Furthermore, a plausible reaction pathway is proposed based on the STM/AFM examination of the reaction intermediates as well as DFT modeling.

The findings and elegant approach introduce an important coupling reaction to the growing field of on-surface chemistry, which opens new avenues towards the synthesis of low-dimensional materials incorporating $sp+sp^2$ hybridized carbons without employing alkyne derivatives. The ms is overall well written, is based on high-quality experimental data, an insightful analysis and touches a timely topic.

Prior to publication a series of points deserves further attention:

1.- regarding carbyne synthesis (see introduction), it is interesting to acknowledge the advances reported in Nature Materials 15, 634–639 (2016) that also take advantage of special environments and reaction conditions

Reply: We appreciate the referee's insightful comments and advice. The paper (*Nature Materials* 15, 634–639 (2016)) reported an interesting route to synthesis of carbyne chains using the confined space inside double-walled carbon nanotubes, which shielded the active carbyne chains from the ambient environments. It is a creative idea towards the bulk production of carbyne chains. So we acknowledged this work in the introduction of the revised manuscript (page 2).

2.- it was pointed out that 'Glaser coupling' is not (always) adequate for interfacial reaction scenarios.

Reply: We agree with the reviewer that although the butadiynylene can be synthesized by applying Glaser coupling, it is difficult to fabricate uniform nanostructures because of the abundant side-reactions. Thus the preparation of extended molecular structures containing ethynylene is still a great challenge and highly desired as a bottom-up approach to constructing nanoscale electronic components. We clarified this point on page 3 of the revised manuscript.

3.- the nature of three-fold node of the branched structure formed at 300 K (Fig.1b and Fig. S1) is not addressed. Similar to the fact that most of the ends of the molecular chains anchor to the step edges, the presence of a Cu atom in the joint connection could indicate the stabilization of carbon radicals via metal atoms (cf. J. Phys. Chem. C 119, 9669 (2015)). The authors shall discuss the possible node structures.

Reply: We updated the Figure S1 in the supplementary materials. The joint point of a three-fold branch appeared as a bright protrusion in STM image (Figure S1a). The corresponding AFM

images (Figure S1b, c) illustrated that the bonded phenylenes tilted toward the substrate, and the joint point showed as a blurred feature. Based on an earlier study,^[1] we attributed the three-fold node to a coordination structure consisting of PPE nanowires and a three-coordinate Cu center. (Figure S1d), the STM and AFM simulations (Figure S1e and f) based on DFT-calculated structures agreed well with the experimental results. The centered Cu adatom and the carbon atoms directly bound to it are lower than the PPE nanowires on Cu(111), thereby cannot be clearly resolved in nc-AFM image.^[2] It is interesting to find that the optimized configuration of Cu adatom is at the top site of Cu(111). The termini of PPE, which are carbyne radicals, bonded to the Cu adatom and affected the local coordination. Thus the Cu adatom located on top site became the most stable structure in this three-fold coordination node. We added the related discussion on *page* 3 of the revised manuscript.

Figure S1. The bonding configuration of three-node branched structures observed in Figure 1b. (a) STM image and (b, c) the corresponding AFM images of branched structure. (d) Top and side view of the DFT-optimized molecular structure of three-fold nodes. (e, f) STM simulation and nc-AFM simulation of three-fold branched structure based on the model in (d).

4.- likewise, what is the chemical structure of the PPE wire termini after 358 K annealing, especially for those resting on the open surface (Fig. 1C)? Is there evidence for surface-stabilized radicals or their passivation by hydrogen atoms? Please include meaningful discussions in the main text.

Reply: The ends of PPE nanowire always appeared as bright features in nc-AFM image (Fig. S3). The earlier studies identified the functional group in similar case as $-\text{CH}_3$,^[3] which was formed by the passivation with hydrogen atoms. Combined with the reply to question 3#, it is suggested that the adatom-coordinated PPE moieties are chemically active. They coupled with each other at the elevated temperature (358 K) and resulted in longer PPE nanowires. The nc-AFM images shown

below were added as supplemental materials (Fig. S3) in the revised manuscript, and the related discussions were added in the main text (on page 3).

Figure S3. An nc-AFM image of PPE nanowire terminus after annealing the sample at 358 K.

5.- in the inset of Fig.1c, there is a vertical displacement between the neighboring wires about half of its periodicity that should be analyzed.

Reply: The high-resolution STM and nc-AFM images of PPE nanowire arrays were shown below. The contrast of the STM image (a) was enhanced so that the dot-like features between neighboring PPE nanowires can be clearly seen. The AFM image (b) revealing the molecular structure clearly showed an axial displacement between neighboring wires. We believe that the dot-like protrusions are Cl atoms detached from precursors typically seen in the coupling reactions using Cl-substituted precursors.^[4,5] Our DFT calculations indicated that the Cl atoms stabilized PPE nanowire arrays by intermolecular Cl...H bonds. The Cl adsorbates were lower in height with respect to the PPE nanowires by ~ 96 pm, thus could be hardly resolved by AFM in constant-height mode. To highlight the importance of this issue, we have updated Figure 2 by adding the STM/AFM images of PPE nanowire arrays (on page 5~7) and also added the DFT calculations and the detailed analysis in supplementary materials (Fig. S7).

(a) STM image and (b) the corresponding AFM image of a segment of the closely packed PPE nanowires. The images were acquired simultaneously in constant-height mode by applying a small tip bias of 2 mV. (c, d) Top view and side view of DFT-optimized model of PPE nanowire array.

6.- it is helpful to show the gas-phase band structure of the PPE wire to have a better understanding of the DOS profile depicted in Fig. 1f.

Reply: We agree with the reviewer. The calculated band structure of PPE nanowire in gas-phase is shown. A significant dispersion was found along the axis of PPE nanowire. The density of states (DOS) of gas-phase PPE nanowire exhibits typical semiconducting feature with two resonance peaks near Fermi level. The energy splitting between the occupied state and the unoccupied state is ~ 1.91 eV, which is close to the bandgap of PPE oligomer in vacuum (~ 1.92 eV). The band structure of the gas-phase PPE nanowire and the discussion is now included in the revised manuscript as supplementary information (Fig. S5).

Figure S5. Calculated band structure along the high symmetry k-path [Γ (0.0 0.0 0.0) \rightarrow Y (0.0 0.5 0.0) \rightarrow A (0.5 0.5 0.0) \rightarrow B (0.5 0.0 0.0) \rightarrow D (0.5 0.0 0.5)] and the corresponding density of states (DOS) of gas-phase PPE nanowire.

7.- the data in fig. 1 also indicate that atomic steps of the substrate can be crossed by reaction products without affecting the geometric order. Indeed, it would be interesting to discuss potential templating routes towards improved control of molecular wires, as demonstrated in the literature.

Reply: After annealing the sample at 358 K, elongated PPE nanowires were found to cross the atomic steps of Cu(111) as seen in STM image (Figure S4a, blue dot lines). We proposed that the PPE nanowires with passivated termini (discussed in question #4) were mobile on Cu(111) surface at 358 K. The polymerization *via* the end-to-end attachment made the PPE wires extended and might overcome the step-edge barrier. The orientation of the nanowires remained unchanged

because of the rigidity of the PPE molecular skeletons and the template effect driven by Cu(111) substrate. In addition, we also observed that PPE nanowire across the atomic steps on Ag(111) (Figure S4b). The above discussion and the STM images were added to the revised manuscript as supplementary material (Fig. S4).

Figure S4. STM images of PPE nanowires on (a) Cu(111) and (b) Ag(111) surface. Blue arrows marked the nanowires across the atomic steps of substrates.

8.- p. 5: although AFM data and simulations roughly agree, significant distortions seem to affect the experimental data. The reasons should be clarified.

Reply: The AFM image (Fig. 2b) showed an elongation of the phenylene rings perpendicular to the axis of PPE nanowire. This effect, which is induced by the tilting effect of CO on tip apex, has been well-studied in nc-AFM community.^[6] The CO relaxation enhanced the resolution of the chemical bonds, with the side effect of imaging distortions. We added the reference and a succinct discussion in the revised manuscript to clarify the issue (on page 6).

9.- p. 5: the carbon-carbon bonding description and (potential) substrate hybridization effects are described in an unconvincing manner. This crucial issue requires a more rigorous treatment.

Reply: We performed comprehensive calculations, including electron localization function (ELF), natural bond orbital (NBO), charge density difference (CDD) and density of states (DOS) of the system of PPE nanowire adsorbed on Cu(111). The optimized atomic structure is shown in Figure S6a. The bond order of the C≡C bonds in PPE nanowire on Cu(111) was calculated to be 2.61, which is very close to that in the gas-phase (2.62). ELF (Figure S6b,c) revealed insignificant electronic hybridization with the underlying Cu substrate, as also confirmed by Bader charge analysis, which showed a small amount of charge transfer from Cu(111) to PPE nanowire by 0.03 e per unit (-Ar-C≡C-). Both ELF and CCD conclude the lack of electronic density, thus insufficient for chemical bonding, between PPE nanowire and underlying Cu(111) surface. Large electron localization in the regime of C≡C bonds in comparison with the adjacent C-C bonds (Figure S6b) explains the high nc-AFM contrast seen on C≡C bonds. Figure S6e further compares the DOS of PPE nanowires in vacuum (upper panel) with that on Cu(111) (lower panel). An overlap between the DOS of PPE and the *d* states of Cu(111) surface was observed in the energy range from -1.0 eV to -3.9 eV. The projected density of states (PDOS) of PPE molecular wire on

Cu(111) suggested a weak hybridization that could be mainly attributed to the interaction between p_z orbitals of PPE nanowire and d_{z^2} orbitals of Cu substrates. We have carefully discussed all of the above points in main text (on page 5, 6). The details of calculation and results were added in supplementary material (Figure S6).

Figure S6 Interaction between PPE nanowire and the underlying Cu(111) surface. (a) the DFT-optimized model of PPE nanowire on Cu(111). (b) Top-view and side-view images of electron localization function (ELF) localization domains (isodensity value is 0.8) for PPE nanowire on Cu(111) surface. (c) Top and side views of ELF maps for PPE on Cu(111). (d) Top and side visualization of charge density difference (CDD) for PPE on Cu(111) ($\Delta\rho = \rho_{\text{tot}} - \rho_{\text{slab}} - \rho_{\text{ads}}$). The isodensity value is 0.0008 e/bohr^3 . Red (blue) denotes positive (negative) regions. The red dash line indicates the location of carbon-carbon triple bond in PPE nanowire. (e) Calculated density of states (DOS) for the PPE nanowire in vacuum (upper panel) and on Cu(111) surface (lower panel) in a larger energy range than that in Figure 1f. The wine-colored line in lower panel represents the d states of Cu(111). (f) Projected density of states (PDOS) of PPE nanowire on Cu(111) surface.

10.- p. 8: the identification of reaction intermediates is partially rather speculative. Although certain similarities between simulated and experimental AFM data exist, the agreement is not generally very good. Discrepancies and potential reasons should be adequately addressed.

Reply: As the reviewer pointed out, the reaction intermediate we proposed cannot be fully proved by our STM and AFM results alone. Although we can routinely acquire bond resolved AFM images on planer molecules, the stand-up species (radicals) and some complex non-planer moieties indeed frustrate the AFM technique. The reaction intermediates in this study were very

likely nonplanar, hindering us to achieve submolecular resolution AFM images that enable straightforward interpretations. To identify the intermediates, we did a careful statistical analysis of large-scale images and a number of AFM images of surface-stabilized species. For example, most close-up AFM images of benzyl radicals reveal similar features of a tilted benzene ring and two neighboring protrusions. According to the calculated model in Fig. 4j, the symmetry axis of surface-stabilized benzyl radicals is along $[11\bar{2}]$ and the equivalent directions of Cu(111) surface. Thus there are six optimal adsorption sites, which were illustrated in diagram below. We found that all the intermediates **I** recorded in AFM images agreed with the adsorption sites predicted by DFT calculations. The reaction intermediates we proposed could be also rationalized by the homocoupling of two 1, 1, 1-trichloromethylaryl in solution,^[7] which yielded the same species that have been well-characterized by means of various spectroscopies. The above discussion and experimental results were added as supplementary material (Fig. S10) to substantiate the identification of reaction intermediates in main text.

Figure S8. The six equivalent adsorption sites of benzyl radicals stabilized by Cu(111) surface and the AFM images of the adsorbates in the corresponding cases.

11.- the experimental findings to support the proposed reaction pathway are not properly presented. When referring to the reaction intermediates, large-scale STM overview images of samples subject to different annealing temperatures are not shown. Statistics of the distribution of intermediates formed at different temperatures shall be provided. Alternatively, a stepwise tip-induced dechlorination experiment can be performed to verify the geometrical changes of the precursor upon activation.

Reply: Thanks for the suggestion. We conducted the annealing experiment to determine the intermediates distribution and evolution in the reaction. The large-scale STM images and close-up STM/AFM characterization of the samples subjected to different annealing temperatures were

displayed in Figure S9. Figure S9a was recorded after TCMB deposited on Cu(111) at 4.7 K. The species marked by green dashed circles could be identified as benzyl radicals by the close-up STM and AFM images, i.e. Fig. S9b,c show the magnified area of left top corner in Fig. S9a; Fig. S9d,e show an molecular cluster formed by four surface-stabilized benzyl radicals. The statistical analysis indicated approximately 90 % of the molecular species formed on Cu(111) at 4.7 K were benzyl radicals (Fig. S9l). After annealing the sample to ~ 50 K, a mixture of benzyl and carbene radicals was observed as shown in Figure S9f. Carbene radicals were marked by yellow dashed oval. Figure S9g, h are close-up STM and AFM images of an island formed by three surface-stabilized carbene radicals. Nearly 70% of the species were carbene radicals at 50 K, while the left 30 % are benzyl radicals (Fig. S9l). An annealing above 70 K (in the range of 70 ~ 200 K) led to a decreased coverage of adsorbates on Cu(111), and only the surface-bound carbyne radicals were observed (marked by blue dashed circles in large-scale STM image Fig. S9i). The necklace-like dots decorating the Cu steps in Figure S9a,i were the detached Cl atoms. The protrusions marked by purple dashed circles in Figure S9c,e were the adsorbed CO molecules. The reaction pathway could be justified by the above-mentioned observations. We added the statistical analysis and discussion in the main text on *page 8*. The experimental results were added in the supplementary materials (Fig. S9).

Figure S9. The overview and statistics of reaction intermediates observed on Cu(111) after the annealing at different temperatures. (a) Large-scale STM image after TCMB deposited on Cu(111) at 4.7 K. (b, c) Close-up STM image and the corresponding AFM image of the region marked in (a) by green dashed (top left corner). (d, e) STM and AFM images of an island formed by four surface-stabilized benzyl radicals. (f) STM image of sample after annealing at ~ 50 K. (g, h) Close-up STM and AFM images of an island formed by three surface-stabilized carbene radicals. (i) Large-scale STM image after the sample annealing at >70 K. (j, k) STM and AFM images of surface-supported carbyne radicals. (l) Statistics for distribution of intermediates formed at different temperatures.

12.- is it expected that this reaction scheme also would work on noble metal or other surfaces?
This is an important point regarding broad employability of the approach.

Reply: Thanks for pointing out this important issue. We did explore the reaction scheme on other noble metal surfaces. We found that the PPE nanowires can be successfully synthesized on Ag(111) (Fig. S13). Additionally, we successfully obtained molecular motifs containing C \equiv C bonds on Au using Br-substituted precursors [just accepted by *Angew. Chem. Int. Ed.* doi:10.1002/ange.201801056]. This information has been added as the latest experimental results in the supplementary materials (Fig. S13).

Figure S13 STM images of PPE nanowires synthesized on Ag(111) surface.

13.- there are repeated spelling / grammatical mistakes, careful proofreading is necessary.

Reply: The spelling and grammatical mistakes have been corrected in the revision.

Reference:

1. Gutzler, R. *et al.* Surface mediated synthesis of 2D covalent organic frameworks: 1,3,5-tris(4-bromophenyl)benzene on graphite(001), Cu(111), and Ag(110). *Chem. Commun.* **45**, 4456-4458 (2009).
2. Schuler, B. *et al.* Adsorption Geometry Determination of Single Molecules by Atomic Force Microscopy. *Phys. Rev. Lett.* **111**, 106103 (2013).
3. Schuler, B. *et al.* Unraveling the Molecular Structures of Asphaltenes by Atomic Force Microscopy. *J. Am. Chem. Soc.* **137**, 9870-9876 (2015).
4. Wang, W. *et al.* Single-Molecule Resolution of an Organometallic Intermediate in a Surface-Supported Ullmann Coupling Reaction. *J. Am. Chem. Soc.* **133**, 13264-13267 (2011).
5. Park, J. *et al.* Interchain Interactions Mediated by Br Adsorbates in Arrays of Metal-Organic Hybrid Chains on Ag(111). *J. Phys. Chem. C.* **115**, 14834-14838 (2011).
6. Moll, N. *et al.* Image Distortions of a Partially Fluorinated Hydrocarbon Molecule in Atomic Force Microscopy with Carbon Monoxide Terminated Tips. *Nano Lett.* **14**, 6127-6131 (2014).
7. Kokil, A. *et al.* High Charge Carrier Mobility in Conjugated Organometallic Polymer Networks. *J. Am. Chem. Soc.* **124**, 9978-9979 (2002).

REVIEWERS' COMMENTS:

Reviewer #2 (Remarks to the Author):

The revision is pertinent and improved the paper substantially.

For completeness, a recent publication aiming at related molecular wire formations incorporating C-C triple bonds should be mentioned (see <https://doi.org/10.1002/sml.201704321>).

Reply to reviewers' comments

Reviewer #2 (Remarks to the Author):

The revision is pertinent and improved the paper substantially. For completeness, a recent publication aiming at related molecular wire formations incorporating C-C triple bonds should be mentioned (see <https://doi.org/10.1002/sml.201704321>).

Reply: Following the reviewer's advice, we acknowledged the paper (doi.org/10.1002/sml.201704321) in the revised manuscript (*on page 2, Ref. 26*).